# Pooled Analysis of the Effect of Pre-Existing Ad5 Neutralizing Antibodies on the Immunogenicity of Adenovirus Type 5 Vector-Based COVID-19 Vaccine from Eight Clinical Trials

**DOI:** 10.3390/vaccines13030333

**Published:** 2025-03-20

**Authors:** Wenqing Liu, Yuqing Li, Xiaolong Li, Feiyu Wang, Runjie Qi, Tao Zhu, Jingxin Li

**Affiliations:** 1School of Public Health, National Vaccine Innovation Platform, Nanjing Medical University, Nanjing 211166, China; lwq19990611@163.com (W.L.); lyq19991023@163.com (Y.L.); qrj1232022@163.com (R.Q.); 2CanSino Biologics Inc., Tianjin 300457, China; xiaolong.li@cansinotech.com (X.L.); feiyu.wang@cansinotech.com (F.W.); 3Jiangsu Provincial Medical Innovation Center, National Health Commission Key Laboratory of Enteric Pathogenic Microbiology, Jiangsu Provincial Center for Disease Control and Prevention, Jiangsu Provincial Academy of Preventive Medicine, Nanjing 210009, China

**Keywords:** COVID-19 vaccine, Ad5-nCoV, pre-existing Ad5 immunity

## Abstract

**Background**: Pre-existing adenovirus immunity restricts the utilization of adenovirus-vectored vaccines. The current study aims to conduct a pooled analysis of eight clinical trials to evaluate the influence of pre-existing Ad5 neutralizing antibodies on immunogenicity of Ad5-nCoV. **Methods**: The primary outcome indicator of this pooled analysis is the geometric mean titers (GMTs) of live SARS-CoV-2 NAbs against the wild-type strain on day 28 post-vaccination. Participants were divided into two cohorts: an adolescent cohort comprising individuals aged 6–17 years and an adult cohort with individuals aged 18 years and older. Within each cohort, individuals were further categorized into three subgroups based on their Ad5-nCoV vaccination schedules: one subgroup received a single intramuscular dose as the primary regimen (Ad5-IM-prime), another received an intramuscular dose as the heterologous prime-boost regimen (Ad5-IM-boost), and the last subgroup received an aerosolized dose as the heterologous prime-boost regimen (Ad5-IH-boost). **Results**: A total of 3512 participants were included in this pooled analysis. In the Ad5-IM-prime subgroup, there were 1001 adolescents and 1450 adults; in the Ad5-IM-boost subgroup, there were 65 adolescents and 396 adults; and in the Ad5-IH-boost subgroup, there were 207 adolescents and 393 adults. In the adult cohort, the GMTs of NAbs against wild-type SARS-CoV-2 on day 28 post-vaccination for the Ad5-IM-prime, Ad5-IM-boost, and Ad5-IH-boost subgroups were 35.6 (95% CI: 32.0, 39.7), 358.3 (95% CI: 267.6, 479.6), and 2414.1 (95% CI: 2006.9, 2904.0), respectively, with negative (less than 1:12) pre-existing NAb titers compared to 10.7 (95% CI: 9.1, 12.6), 116.9 (95% CI: 84.9, 161.1), and 762.7 (95% CI: 596.2, 975.8), respectively, with high (greater than 1:1000) pre-existing NAb titers. A similar trend was observed in the adolescent cohort, where pre-existing immunity was found to reduce the peak of live SARS-CoV-2 Nabs post-vaccination. **Conclusions**: Regardless of whether Ad5-nCoV is administered as a primary vaccination regimen or as a heterologous prime-boost strategy, a negative impact on immunogenicity can still be observed in the presence of high pre-existing immunity. However, when primary immunization is achieved with inactivated COVID-19 vaccines, aerosol inhalation can significantly enhance the immunogenicity of Ad5-nCoV compared to intramuscular injections of Ad5-nCoV as a booster.

## 1. Introduction

Adenoviruses are non-enveloped viruses with a double-stranded DNA genome, comprising 252 capsomers arranged in an icosahedral structure. Among these, 240 are hexons and 12 are penton bases. Each hexon is a homotrimer of hexon proteins, and the epitopes on hexons are standard for diagnosing different serotypes. Each penton base is associated with a fiber protrusion, which has serotype specificity and contains specific antigen binding sites for agglutination of cells in vitro [1]. Adenoviruses themselves have a high immunogenicity and can strongly induce neutralizing responses against the virus’s surface hexon, penton proteins, and fibers after infection.

Adenoviral (Ad) vectors are considered a promising candidate for an ideal vaccine vector due to their fulfilling key criteria related to efficacy, safety, and stability [2]. These vectors are particularly effective at stimulating a robust innate immune response, which is crucial for subsequently promoting elevated levels of both humoral and cellular immune responses. This phenomenon results largely from their ability to preferentially target antigen-presenting cells, which are essential for initiating broad immune reactions [3]. Several recombinant Ad-vectored vaccines have successfully demonstrated their capacity to elicit both humoral and cellular immune responses in various animal models and human clinical trials. Notably, studies conducted in mice [4], non-human primates [5,6], and human participants [7] have yielded encouraging results, highlighting the effectiveness of these vaccines in generating the desired immune responses. Over the past few decades, this innovative vaccine platform has been extensively investigated in both preclinical and clinical settings, leading to the development of vaccines aimed at combating a range of viruses responsible for infectious diseases. Significant examples include vaccines designed to target the Ebola virus [8,9], the Zika virus [10], and the human immunodeficiency virus [11], which demonstrate the versatility and potential applications of Ad vectors in vaccine research and development.

However, pre-existing anti-Ad5 immunity is considered to be the biggest obstacle that candidate Ad5-vectored vaccines must overcome. Studies indicate that the seroprevalence of Ad5 in healthy populations in China is approximately 60% to 82% [12,13,14], while in some sub-Saharan African countries, it may be as high as 100% [15]. When the Ad5 vector vaccine is administered, these pre-existing antibodies can neutralize the vector virus, preventing it from infecting cells and limiting the expression of the antigen gene it carries [16]. This neutralization weakens the level of the specific immune response induced by the vaccine and accelerates the decline of antibody titers, ultimately affecting the vaccine’s immunogenicity.

Despite the widespread pre-existing immunity in humans, Ad5 is still widely regarded as one of the most typical vector systems. During the severe acute respiratory syndrome coronavirus 2 (SARS-CoV-2) pandemic, CanSino Biologics Inc. and the Beijing Institute of Biotechnology developed a replication-defective recombinant Ad5 vaccine (Ad5-nCoV) utilizing an E1/E3 deleted, replication-defective Ad5 encoding the full-length, mammalian-expression-optimized Spike gene with a tissue plasminogen activator (tPA) signal peptide [16]. Here, we conduct a pooled analysis of eight clinical trials to assess the impact of pre-existing Ad5 neutralizing antibodies (NAbs) on live SARS-CoV-2 neutralizing antibodies (NAbs) after receiving Ad5-nCoV as either the primary series vaccination or as part of a heterologous prime-boost regimen.

## 2. Methods

### 2.1. Study Design and Data Source

We conducted this pooled analysis based on eight completed clinical trials involving Ad5-nCoV (Appendix A), such trials being primarily divided into two categories. The first category encompassed three primary immunization trials that involved a single intramuscular injection of Ad5-nCoV containing 3.0 × 10^10^, 5.0 × 10^10^, 1.0 × 10^11^, or 1.5 × 10^11^ viral particles in healthy individuals aged 6 years and older. The second category comprised five heterologous prime-boost immunization trials. In these trials, aerosolized Ad5-nCoV containing 1.0 × 10^10^ viral particles or intramuscular Ad5-nCoV containing 3.0 × 10^10^ viral particles was administered as a third dose to healthy children and adolescents aged 6 to 17 years who had previously received two doses of an inactivated COVID-19 vaccine as their primary immunization. Additionally, aerosolized Ad5-nCoV containing either 1.0 × 10^10^ or 2.0 × 10^10^ viral particles, or intramuscular Ad5-nCoV containing 5.0 × 10^10^ viral particles, was administered as a second, third, or fourth dose to healthy adults aged 18 years or older who had completed one, two, or three doses of vaccination with an inactivated COVID-19 vaccine. We collected and analyzed data primarily including participants’ age, sex, dosage received, pre-existing Ad5 neutralizing antibody (NAb) titers on day 0 pre-vaccination, as well as live SARS-CoV-2 NAb titers both on day 0 pre-vaccination and on day 28 post-vaccination. The specific design and assay protocols of each clinical trial have been fully described in previous publications [17,18,19,20,21].

### 2.2. Antibody Detection

In all participants included in the eight clinical trials, the detection of pre-existing Ad5 NAbs on day 0 pre-vaccination was based on the firefly luciferase assay system. Serum samples from 3512 participants were subjected to three-fold serial dilutions, ranging from 1:12 to 1:8748, and duplicate sets were prepared for each dilution concentration. To establish a baseline for comparison, a diluent that contained no serum was utilized as a negative control in this experimental setup. In the wells designated for the positive control, serum was excluded entirely, leading to the highest level of luciferase activity observed. This maximum activity was crucial for the precise calculation of 90% neutralization values. Ad5-luciferase was mixed with an equal volume of each diluted serum sample and incubated for 1 h in a 96-well plate, allowing for sufficient interaction [22]. Following this incubation period, a suspension of A549 cells, obtained from the American Strain Preservation Center, was introduced to the mixture. After a 24 h incubation period at 37 °C, the cells underwent a washing procedure, and then they were lysed to facilitate the measurement of luciferase activity. The luciferase measurements were carried out using the Firefly Luciferase Assay system produced by Promega, and the resulting values were quantified utilizing the GloMax Microplate luminometer, also from Promega.

In addition, the levels of live SARS-CoV-2 NAbs against the wild-type strain in the serum samples from all participants were determined on day 0 pre-vaccination and on day 28 post-vaccination with Ad5-nCoV utilizing a microneutralization assay based on cytopathic effects. This involved preparing serum dilutions which were subsequently combined with an equal volume of viral solution, yielding a final viral concentration of 100 TCID50 per well. The titers of the NAbs were expressed as the reciprocal values of the highest serum dilutions that succeeded in protecting at least 50% of cultured cells from observable cytopathic effects caused by the virus. The dilution series for the microneutralization assay commenced at a ratio of either 1:4 or 1:8, ensuring a methodical approach to accurately assess the serum’s neutralizing capabilities against the wild-type strain.

### 2.3. Statistical Analysis

We selected the per-protocol set (PPS) of participants from these eight clinical trials, totaling 3512 participants, all of whom were vaccinated with Ad5-nCoV. The primary outcome indicator of the study was the geometric mean titer (GMT) of live SARS-CoV-2 NAbs against the wild-type strain on day 28 post-vaccination. In this study, participants were initially divided into two primary age-based cohorts: adolescents aged 6 to 17 and adults aged 18 and older. Within each cohort, individuals were further categorized into three distinct subgroups based on their Ad5-nCoV vaccination schedules: the first subgroup received a single dose via intramuscular injection as the primary regimen (Ad5-IM-prime), the second subgroup received a single dose via intramuscular injection as the heterologous prime-boost regimen (Ad5-IM-boost), and the third subgroup received a single dose via aerosolized inhalation as the heterologous prime-boost regimen (Ad5-IH-boost). It is noteworthy that participants in the Ad5-IM-boost and Ad5-IH-boost subgroups had previously been primed with one or two doses of the inactivated COVID-19 vaccine or had completed homologous booster immunization with three doses of the inactivated COVID-19 vaccine prior to receiving Ad5-nCoV as a heterologous booster. Data from these trials were meticulously categorized and merged according to the age of the participants, the immunization schedule, and the dosage of the vaccination (Table 1).

Data analysis in this study included the following steps. First, stratified analyses of the immune responses were conducted based on pre-existing Ad5 NAb titers on day 0 before vaccination, categorizing participants as negative (less than 1:12), low (1:12 to 1:200), moderate (1:200 to 1:1000), or high (greater than 1:1000). Second, we constructed multiple linear regression models utilizing data from eight clinical trials. In each model, the independent variable was consistently the level of pre-existing Ad5 NAbs, while the dependent variable was the level of live SARS-CoV-2 NAbs on day 28 post-vaccination. We also considered covariates such as age, sex, different immunization schedules or doses, live SARS-CoV-2 NAbs on day 0 pre-vaccination, and the time since the last dose of the inactivated COVID-19 vaccine to assess their impact on the models. In model 1, pre-existing Ad5 NAb titers were treated as categorical variables, while in model 2, they were analyzed as continuous variables. Third, we compared the geometric mean titers (GMTs), geometric mean fold increase (GMFI), and the four-fold increase in titers on day 28 post-vaccination among the three subgroups within both adolescent and adult cohorts.

Continuous data are presented either as the mean with SD or as the median with IQR, while categorical data are presented as percentages. The titers of antibodies were transformed using a logarithmic scale prior to conducting any statistical analyses and are reported as geometric mean titers (GMTs). Categorical data were analyzed using either the chi-squared (χ^2^) test or Fisher’s exact test. Additionally, analysis of variance (ANOVA) was employed to evaluate the log-transformed antibody titers among the three subgroups with different vaccination regimens, and the Student–Newman–Keuls test was performed to conduct multiple comparisons. All analyses were performed in R version 4.3.3, and figures were created with GraphPad Prism version 9.00 (GraphPad Software, San Diego, CA, USA). Two-sided *p* values < 0.05 were considered to be significant.

## 3. Results

### 3.1. Study Population

This pooled data analysis from eight clinical trials included a total of 3512 participants, who were further categorized into 1273 adolescents aged 6 to 17 years and 2239 adults aged 18 years and older. In the adolescent cohort, participants were divided into three different immunization regimen subgroups: Ad5-IM-prime subgroup (n = 1001), Ad5-IM-boost subgroup (n = 65), and Ad5-IH-boost subgroup (n = 207). Pre-existing Ad5 NAb titers observed on day 0 prior to vaccination showed no statistical difference (*p* = 0.3146) among the three subgroups, with geometric mean titers (GMTs) of 118.3 (95% CI: 101.7, 137.6) in the Ad5-IM-prime subgroup, 73.0 (95% CI: 37.1, 143.7) in the Ad5-IM-boost subgroup, and 116.4 (95% CI: 81.5, 166.0) in the Ad5-IH-boost subgroup. However, the distribution of pre-existing Ad5 NAb titers among the three subgroups (Ad5-IM-prime, Ad5-IM-boost, and Ad5-IH-boost) showed significant differences (*p* = 0.0008). In the Ad5-IM-prime subgroup, the proportions of negative (less than 1:12), low (1:12 to 1:200), moderate (1:200 to 1:1000), and high (greater than 1:1000) pre-existing Ad5 NAb titers were 33.6%, 11.4%, 42.4%, and 12.6%, respectively. In the Ad5-IM-boost subgroup, the proportions were 49.2%, 10.8%, 20.0%, and 20.0%, respectively. Meanwhile, the Ad5-IH-boost subgroup exhibited proportions of 36.7%, 10.1%, 33.3%, and 19.8%, respectively.

In the adult cohort, participants were also divided into three different immunization regimen subgroups: Ad5-IM-prime subgroup (n = 1450), Ad5-IM-boost subgroup (n = 396), and Ad5-IH-boost subgroup (n = 393). Pre-existing Ad5 NAb titers observed on day 0 prior to vaccination showed a statistical difference (*p* = 0.0004) among the three subgroups. GMTs of NAbs against wild-type SARS-CoV-2 were higher in the Ad5-IH-boost subgroup [236.8 (95% CI:189.9, 295.1)] compared to the Ad5-IM-prime subgroup [146.5 (95% CI: 131.5, 163.3)] and the Ad5-IM-boost subgroup [165.7 (95% CI: 134.8, 203.8)]. Specifically, the distribution of pre-existing Ad5 NAb titers among the three subgroups (Ad5-IM-prime, Ad5-IM-boost, and Ad5-IH-boost) showed a significant difference (*p* = 0.0026). In the Ad5-IM-prime subgroup, the proportions of negative (less than 1:12), low (1:12 to 1:200), moderate (1:200 to 1:1000), and high (greater than 1:1000) pre-existing Ad5 NAb titers were 20.5%, 25.1%, 40.9%, and 13.8%, respectively. In the Ad5-IM-boost subgroup, the proportions were 16.7%, 27.5%, 44.2%, and 11.6%, respectively. Meanwhile, the Ad5-IH-boost subgroup exhibited proportions of 15.0%, 23.9%, 40.7%, and 20.4%, respectively. Other demographic characteristics are reported in Table 2.

### 3.2. Impact of Pre-Existing Ad5 NAbs on Live SARS-CoV-2 NAbs After Receiving Ad5-nCoV as a Primary Immunization Strategy

In both the adolescent cohort (ages 6–17 years) and the adult cohort (ages 18 years and older), neutralizing antibody titers against wild-type SARS-CoV-2 in the Ad5-IM-prime subgroup were moderate on day 28 post-vaccination. Specifically, the two cohorts exhibited GMTs of 18.5 (95% CI: 17.3, 19.9) for adolescents and 13.9 (95% CI: 13.1, 14.7) for adults, with GMFIs of 9.3 (8.6, 9.9) and 5.5 (5.2, 5.9), respectively. The seroconversion rates were 86.2% for adolescents and 67.4% for adults (Appendix A). Based on a stratified analysis of the Ad5-IM-prime subgroup population, serum NAb titers against the wild-type isolate on day 28 post-vaccination showed a gradual decline in the adolescent cohort as pre-existing NAb titers increased, with GMTs of 36.5 (95% CI: 33.0, 40.4), 25.4 (95% CI: 21.9, 29.4), 12.0 (95% CI: 10.9, 13.2), and 9.9 (95% CI: 8.3, 11.9) for negative (less than 1:12), low (1:12 to 1:200), moderate (1:200 to 1:1000), and high (greater than 1:1000) pre-existing NAb titers, respectively. A similar trend was observed in the adult cohort, with GMTs of 35.6 (95% CI: 32.0, 39.7), 14.1 (95% CI: 12.6, 15.6), 9.4 (95% CI: 8.6, 10.2), and 10.7 (95% CI: 9.1, 12.6) for negative (less than 1:12), low (1:12 to 1:200), moderate (1:200 to 1:1000), and high (greater than 1:1000) pre-existing NAb titers, respectively (Figure 1).

Multiple linear regression analyses were conducted on the Ad5-IM-prime subgroup in the adolescent cohort (Appendix A). In model 1, when using negative pre-existing NAb titers as the reference, the regression coefficients for the impact of low, moderate, and high pre-existing NAb titers on live SARS-CoV-2 NAb titers post-vaccination were −1.4 (*p* = 0.0009), −2.9 (*p* < 0.0001), and −3.5 (*p* < 0.0001), respectively. This indicates that higher levels of pre-existing Ad5 NAbs have a greater negative impact on live SARS-CoV-2 NAb levels on day 28 post-vaccination. In model 2, the results indicate that the regression coefficient for pre-existing Ad5 NAbs was −1.6 (*p* < 0.0001), implying a decrease of 1.6 units in post-vaccination antibody levels for each unit increase in pre-existing Ad5 NAbs. Additionally, the effects of the age and sex variables on post-vaccination antibody levels were found to be negative, with estimated values of −1.0 (*p* < 0.0001) for age and −1.1 (*p* = 0.0350) for sex, with male sex serving as the reference in model 2.

In line with the findings from the adolescent cohort, the multiple linear regression analyses of the Ad5-IM-prime subgroup in the adult cohort reveal that, whether in model 1 or model 2, higher pre-existing Ad5 NAb titers were associated with a more pronounced negative effect on post-vaccination live SARS-CoV-2 NAb levels (Appendix A). In model 1, when using the negative pre-existing NAb titers as the reference, the regression coefficients for the impact of low, moderate, and high pre-existing NAb titers on live SARS-CoV-2 NAb titers post-vaccination were −2.1 (*p* < 0.0001), −3.4 (*p* < 0.0001), and −3.5 (*p* < 0.0001), respectively. Simultaneously, in model 2, when pre-existing NAb titers were treated as a continuous variable, the regression coefficient was −1.7 (*p* < 0.0001). This finding indicates that, for each unit increase in pre-existing Ad5 NAbs, there is a corresponding decrease of 1.7 units in post-vaccination antibody levels. Compared to the high dosage (1.5 × 10^11^ vp) in model 2, the regression coefficients for the medium dosage (1.0 × 10^11^ vp) and low dosage (5.0 × 10^10^ vp) were −1.6 (*p* = 0.0104) and −2.2 (*p* < 0.0001), respectively. Furthermore, the results from model 2 (estimate= −1.0, *p* < 0.0001) show that age negatively affected live SARS-CoV-2 NAbs on day 28 post-vaccination, consistent with the adolescent cohort, while sex had no significant impact on live SARS-CoV-2 NAbs on day 28, a finding which is inconsistent with the adolescent cohort.

### 3.3. Impact of Pre-Existing Ad5 NAbs on Live SARS-CoV-2 NAbs After Receiving Ad5-nCoV as a Heterologous Boosting Immunization Strategy

Participants who received heterologous booster vaccinations of Ad5-nCoV administered via either intramuscular injection or aerosolized inhalation experienced a rapid increase in their antibody titers. In the adolescent cohort, 28 days after booster vaccination, the GMTs of NAbs against wild-type SARS-CoV-2 were 349.3 (95% CI: 290.8, 419.5) for the Ad5-IM-boost subgroup and 461.1 (95% CI: 400.9, 530.3) for the Ad5-IH-boost subgroup, with GMFIs of 83.5 (95% CI: 69.7, 100.1) and 102.8 (95% CI: 88.9, 118.8), respectively, showing no statistically significant differences between the two subgroups (Appendix A). However, the serum GMTs of NAbs against wild-type SARS-CoV-2 on day 28 post-vaccination in the adult cohort were significantly higher in the Ad5-IH-boost subgroup 1102.7 (95% CI: 982.6, 1237.5) compared with the Ad5-IM-boost subgroup [194.5 (95% CI: 172.0, 219.9), *p* < 0.0001]. The GMFIs of NAbs against wild-type SARS-CoV-2 increased to 210.8 (95% CI: 180.0, 246.9) in the Ad5-IH-boost subgroup compared with 52.2 (95% CI: 46.4, 58.8) in the Ad5-IM-boost subgroup at the same timepoint (*p* < 0.0001). In both the adolescent and adult cohorts, the seroconversion rates in the two heterologous sequential booster groups (Ad5-IM-boost and Ad5-IH-boost subgroups) exceeded 98%. Furthermore, similar to the findings in the Ad5-IM-prime subgroup, high levels of baseline pre-existing Ad5 NAb titers were found to reduce the peak of post-vaccination live SARS-CoV-2 NAbs in both the Ad5-IM-boost and Ad5-IH-boost subgroups. Specifically, in the Ad5-IH-boost subgroup of the adult cohort, GMTs of NAbs against wild-type SARS-CoV-2 on day 28 post-vaccination were 2414.1 (95% CI: 2006.9, 2904.0), 1458.9 (95% CI: 1182.5, 1799.8), 842.5 (95% CI: 695.4, 1020.8), and 762.7 (95% CI: 596.2, 975.8) for participants with negative (less than 1:12), low (1:12 to 1:200), moderate (1:200 to 1:1000), and high (greater than 1:1000) pre-existing NAb titers, respectively (Figure 1).

Multiple linear regression analyses were conducted on the Ad5-IM-boost subgroup and Ad5-IH-boost subgroup in the adolescent cohort (Appendix A). In model 1 and model 2, the impact of age, sex, and live SARS-CoV-2 NAbs on day 0 pre-vaccination and time since the last dose of inactivated vaccine on post-vaccination antibody levels in the sequential booster subgroup was not statistically significant (*p* > 0.05). Consistent with the analysis from the Ad5-IM-prime subgroup, the higher the pre-existing Ad5 NAb titers, the greater the impact on post-vaccination antibody levels in model 1. Specifically, using negative pre-existing Ad5 NAb titers as a reference, the regression coefficients for low, moderate, and high pre-existing Ad5 NAb titers were −1.5 (*p* = 0.0168), −2.0 (*p* < 0.0001), and −3.2 (*p* < 0.0001), respectively. Simultaneously, in model 2, when pre-existing NAb titers were treated as a continuous variable, the regression coefficient was −1.5 (*p* < 0.0001). In model 2, compared to the Ad5-IM-boost subgroup, the live SARS-CoV-2 NAb levels of the Ad5-IH-boost subgroup increased by 1.4 units post-vaccination.

Simultaneously, multiple linear regression analyses were conducted on the Ad5-IM-boost subgroup and Ad5-IH-boost subgroup in the adult cohort (Appendix A). In model 1, the Ad5-IH-boost subgroup was observed to exhibit significantly elevated NAb levels post-vaccination, with an estimate of 3.9, when compared to the Ad5-IM-boost subgroup. Furthermore, when examining the influence of pre-existing Ad5 NAb titers, a discernible trend emerged: individuals with low titers showed a negative effect (estimate = −1.7), while those with moderate (estimate = −2.4) and high titers (estimate = −3.0) demonstrated an increasingly pronounced negative impact, using negative pre-existing Ad5 NAb titers as the reference for comparison. In model 2, when pre-existing NAb titers were treated as a continuous variable, the regression coefficient was −1.4 (*p* < 0.0001). This finding indicates that, for each unit increase in pre-existing Ad5 NAbs, there is a corresponding decrease of 1.4 units in log-transformed post-vaccination antibody levels. Additionally, the impacts of age (estimate = −1.0, *p* < 0.0001) and time since the last dose of inactivated vaccine (estimate = 1.0, *p* < 0.0001) on post-vaccination antibody levels in the sequential booster subgroup were found to be statistically significant in model 2.

## 4. Discussion

Our results show that the Ad5-nCoV vaccine indicates significant NAb titers against live SARS-CoV-2 on day 28 post-vaccination, but a negative correlation with the pre-existing Ad5 NAb titers before vaccination, regardless of whether the cohort is composed of adolescents aged 6 to 17 years or adults aged 18 years and older. Specifically, higher pre-existing Ad5 NAb titers have a greater impact on live SARS-CoV-2 NAbs against the wild-type strain on day 28 post-vaccination, in line with previous reports [16,23,24]. A study has reported that the overall proportion of seropositive healthy adults in China is 72%, with geographical differences and climate identified as the primary factors influencing the levels of Ad5 titers among healthy adults [13]. In our study, among the 1273 children and adolescents assessed, the seropositivity rate for Ad5 NAbs was 65%, while the seropositivity rate for Ad5 NAbs among the 2239 adults evaluated was 80%. These findings are consistent with previous reports [12,14,15]. Human adenovirus type 5 is widely prevalent in the population, resulting in a majority of individuals possessing pre-existing immunity against Ad5.

Overall, our analyses found that booster vaccinations significantly enhanced the serum neutralizing antibody response against SARS-CoV-2 in adults and adolescents who had previously received one, two, or three doses of an inactivated COVID-19 vaccine, with heterologous booster immunization using aerosolized Ad5-nCoV demonstrating particularly strong immunogenicity and much higher serum neutralizing antibody responses. In the adolescent cohort, after adjusting for factors such as pre-existing Ad5 NAb titers, age, sex, the time since the last dose of the inactivated COVID-19 vaccine, and live SARS-CoV-2 NAbs on day 0 pre-vaccination, the results of the multiple regression analysis indicated that the Ad5-IH-boost subgroup had an average increase of 1.4 units in the level of live SARS-CoV-2 NAbs on day 28 post-vaccination compared to the Ad5-IM-boost subgroup. In the adult cohort, the regression analysis results showed that the Ad5-IH-boost subgroup had an average increase of 3.9 units in the level of live SARS-CoV-2 NAbs on day 28 post-vaccination compared to the Ad5-IM-boost subgroup. In the Ad5-IM-prime subgroup, age consistently had a negative impact on NAb titers against live SARS-CoV-2 post-vaccination, regardless of whether the individuals were aged 6–17 or 18 and above. However, when a heterologous booster immunization strategy was implemented, specifically in individuals within the Ad5-IM-boost subgroup and Ad5-IH-boost subgroup, age no longer had a significant impact on antibody levels post-vaccination.

Due to prior natural infection or vaccination, individuals typically possess a pre-existing immune response to the Ad5 vector. These pre-existing Ad5 NAbs may diminish the specific immune responses to vaccination, particularly the humoral immune responses, by neutralizing the viral vector and impacting the efficiency of vaccine particle entry into cells. Prior research has indicated that both heterologous and homologous prime-boost strategies employing Ad5-vectored vaccines are capable of inducing more robust and long-lasting immunogenic responses in populations with elevated pre-existing immunity to the Ad5 vector [9,25,26]. Nevertheless, for homologous prime-boost regimens, it is crucial to note that pre-existing Ad5 NAbs may diminish the vaccine’s immunogenicity. To mitigate the negative effects of these pre-existing Ad5 NAbs, it is recommended to adopt a longer prime-boost interval. Furthermore, a study has demonstrated that the ChAdOx1 nCoV-19 vaccine offers significantly improved protective efficacy when administered with a longer prime-boost interval (≥12 weeks) compared to a shorter interval (<6 weeks) [27].

A study found that administering the AdHu5Ag85A vaccine via intramuscular injection increases the titers of pre-existing Ad5 NAbs in circulation, whereas vaccination through aerosolized inhalation does not enhance the titers of pre-existing Ad5 NAbs in either the airways or circulation [28]. Delivering Ad-vectored vaccines via the respiratory mucosal route in humans can help bypass pre-existing anti-vector immunity, which tends to be more prevalent in circulation than in the lungs^31^. Furthermore, some studies have shown that respiratory mucosal immunity is more effective than intramuscular immunity in inducing neutralizing antibodies, mucosal tissue-resident memory T cells, and trained airway macrophages [29,30,31,32]. Although oral aerosolized Ad5-nCoV can significantly enhance immunogenicity using a heterologous prime-boost regime, our study still shows in the stratified analysis of pre-existing Ad5 NAb levels that high pre-existing immunity has an inhibitory effect on the levels of live SARS-CoV-2 NAbs post-vaccination.

There are several limitations to our study. First of all, we did not evaluate the antibody levels against other SARS-CoV-2 variants. Second, we did not report the elicited mucosal immunities in individuals who received aerosolized Ad5-nCoV. Third, the sample size of the Ad5-IM-boost subgroup and the Ad5-IH-boost subgroup was relatively small. The relationship between the different routes of Ad5-nCoV administration and the pre-existing Ad5 NAbs titers indeed requires more data to be further clarified. Fourth, in this study, we only evaluated the effect of pre-existing Ad5 NAb on SARS-CoV-2 NAb levels after immunization, but the more complicated pre-existing Ad5 immunity on vaccine-induced T-cell responses was not clear.

## 5. Conclusions

In conclusion, regardless of whether Ad5-nCoV is administered as a primary vaccination regimen or as a heterologous prime-boost strategy, a negative impact on immunogenicity can still be observed in the presence of high pre-existing immunity. However, when primary immunization is achieved with inactivated vaccines, aerosol inhalation can significantly enhance the immunogenicity of Ad5-nCoV compared to intramuscular injection of Ad5-nCoV as a booster.

## Figures and Tables

**Figure 1 vaccines-13-00333-f001:**
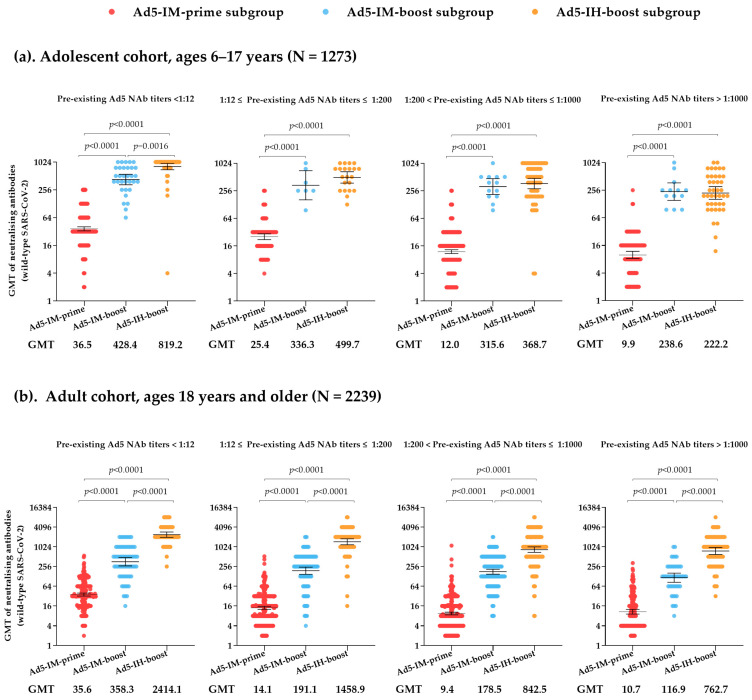
Neutralizing antibodies against live SARS-CoV-2 on day 28 post-vaccination in the adolescent cohort (ages 6–17 years) (**a**) and adult cohort (ages 18 years and older) (**b**), with stratified analysis based on pre-existing Ad5 neutralizing antibodies. GMT = geometric mean antibody titer. Error bars indicate 95% CIs.

**Table 1 vaccines-13-00333-t001:** Data classification of neutralizing antibodies against live SARS-CoV-2 on day 28 post-vaccination for all participants in the eight clinical trials.

Immunization Schedule [Dosage]		6–17 Years ^a^	18 Years or Older ^a^
**Ad5-IM-prime**	Ad5-IM [3.0 × 10^10^ vp]	NCT04916886	/
	Ad5-IM [5.0 × 10^10^ vp]	/	NCT04313127 NCT04341389 NCT04916886
	Ad5-IM [1.0 × 10^11^ vp]	/	NCT04313127 NCT04341389
	Ad5-IM [1.5 × 10^11^ vp]	/	NCT04313127
**Ad5-IM-boost**	ICV × 2 + Ad5-IM [3.0 × 10^10^ vp]	NCT05330871	/
	ICV × 1 + Ad5-IM [5.0 × 10^10^ vp]	/	NCT04892459 NCT04952727
	ICV × 2 + Ad5-IM [5.0 × 10^10^ vp]	/	NCT04892459 NCT04952727
	ICV × 3 + Ad5-IM [5.0 × 10^10^ vp]	/	NCT05303584
**Ad5-IH-boost**	ICV × 2 + Ad5-IH [1.0 × 10^10^ vp]	NCT05330871	NCT05043259
	ICV × 2 + Ad5-IH [2.0 × 10^10^ vp]	/	NCT05043259
	ICV × 3 + Ad5-IH [1.0 × 10^10^ vp]	/	NCT05303584

^a^ All trials in this study are registered with ClinicalTrials.gov. The numbers starting with the characters NCT_ indicate the ClinicalTrials.gov identifier. Abbreviation: vp = viral particles; Ad5-IM= administering Ad5-nCoV by intramuscular injection; Ad5-IH= administering Ad5-nCoV by aerosolized inhalation; ICV = administering inactivated COVID-19 vaccine by intramuscular injection.

**Table 2 vaccines-13-00333-t002:** Baseline characteristics of participants in the eight clinical trials of Ad5-nCoV.

	Ad5-IM-Prime	Ad5-IM-Boost	Ad5-IH-Boost	*p* Value
**Adolescent cohort, ages 6–17 years (N = 1273)**				
**Participants, n**	1001	65	207	/
**Median age (IQR)**	12.0 (10.0, 14.0)	13.0 (9.0, 14.0)	12.0 (10.0, 14.0)	0.6141
**Sex (%)**				
Male	542 (54.1%)	31 (47.7%)	103 (49.8%)	0.3446
Female	459 (45.9%)	34 (52.3%)	104 (50.2%)	
**Pre-existing Ad5 neutralising antibodies**				
Geometric mean titers (95% CI)	118.3 (101.7, 137.6)	73.0 (37.1, 143.7)	116.4 (81.5, 166.0)	0.3146
Participants with titers < 1:12 (%)	336 (33.6%)	32 (49.2%)	76 (36.7%)	0.0008
1:12 ≤ Participants with titers ≤ 1:200 (%)	114 (11.4%)	7 (10.8%)	21 (10.1%)	
1:200 < Participants with titers ≤ 1:1000 (%)	425 (42.4%)	13 (20.0%)	69 (33.3%)	
Participants with titers > 1:1000 (%)	126 (12.6%)	13 (20.0%)	41 (19.8%)	
**Adult cohort, ages 18 years and older (N = 2239)**				
**Participants, n**	1450	396	393	/
**Median age (IQR)**	42.0 (33.0, 50.0)	53.0 (44.0, 65.0)	43.0 (35.0, 50.0)	<0.0001
**Sex (%)**				
Male	749 (51.7%)	215 (54.3%)	171 (43.5%)	0.0048
Female	701 (48.3%)	181 (45.7%)	222 (56.5%)	
**Pre-existing Ad5 neutralising antibodies**				
Geometric mean titers (95% CI)	146.5 (131.5, 163.3)	165.7 (134.8, 203.8)	236.8 (189.9, 295.1)	0.0004
Participants with titers < 1:12 (%)	297 (20.5%)	66 (16.7%)	59 (15.0%)	0.0026
1:12 ≤ Participants with titers ≤ 1:200 (%)	364 (25.1%)	109 (27.5%)	94 (23.9%)	
1:200 < Participants with titers ≤ 1:1000 (%)	593 (40.9%)	175 (44.2%)	160 (40.7%)	
Participants with titers > 1:1000 (%)	200 (13.8%)	46 (11.6%)	80 (20.4%)	

Data are reported as n (%), the median (IQR), or the geometric mean titer (95% CI). The *p* values are the result of comparisons among the three subgroups.

## Data Availability

The data presented in this study are available upon request from the corresponding author.

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
