# Peer review of "Pooled Analysis of the Effect of Pre-Existing Ad5 Neutralizing Antibodies on the Immunogenicity of Adenovirus Type 5 Vector-Based COVID-19 Vaccine from Eight Clinical Trials"

_vaccines, 2025, doi:10.3390/vaccines13030333_

Round 1
Reviewer 1 Report (Previous Reviewer 1)
Comments and Suggestions for Authors
The improved version may be published
Reviewer 2 Report (Previous Reviewer 2)
Comments and Suggestions for Authors
I have no further suggestions for improvement of the manuscript.
This manuscript is a resubmission of an earlier submission. The following is a list of the peer review reports and author responses from that submission.
Round 1
Reviewer 1 Report
Comments and Suggestions for Authors
The manuscript is devoted to the study of influence of pre-existing Ad5-neutralizing antibodies on the immunogenicity of Ad5 vector-based COVID-19 vaccine in different ways of inoculation from eight clinical trials made on adolescent and adult volunteers. The results are valuable and will be useful to every researcher who is developing adenovirus-based vector vaccines. It is very useful that the authors presented the special paragraph in the end of Discussion chapter explaining the limitations of the study. The main part of Conclusion is well-proven by experimental results but there are some issues which should be corrected. The comments are presented below.
1. The abstract should be well understandable without reading the whole text. But in this manuscript it is not clear from the abstract what “heterologous regimen” means. It became clear only from the main text. In addition, the abstract contains the GMT abbreviation which meaning became clear only from the text, too, because it usually means Greenwich Mean Time. The last sentence in the abstract is not well understandable and should be rewritten in more clear way.
2. The description of the different volunteer’s groups is too complicated and requires reading the text a few times. It would be much clear, if the special table with groups immunization description and its schedules was made in the beginning of Results chapter or in the Methods section.
3. The last sentence in Conclusion paragraph is, again, not clear because it is written in the same manner as the last sentence of the Abstract.
The corrected version should be presented for reviewer’s evaluation.
Reviewer 2 Report
Comments and Suggestions for Authors
The paper shows a potential problem with replication defective Ad5-linked CoV-2 vaccines. Because many humans have been exposed to Ad5 virus, pre-existing antibodies against the Ad5 portion of the vaccine may inhibit the effectiveness of the subsequent inoculation with the Ad5-linked CoV-2 spike protein. The paper should make clear what the potential benefits of the use of an Ad5-linked CoV-2 protein vaccine might be and why this system is being used in current clinical trials. This reviewer is not very familiar with the Ad5 system, so, perhaps, this reviewer should know this already. I assume that the replication-defective virus is an effective means of antigen delivery to induce a robust immune response, but I would like this, or whatever the case may be, to be explicitly stated. Of course, there are many alternate means of vaccination against CoV-2 and these approaches may be preferable to the Ad5-linked CoV-2 vaccine.
The pooled clinical trials tested the Ad5-CoV-2 vaccine in many people. The data presented are comprehensive and reasonably presented.
The paper is well-written and presented.
Reviewer 3 Report
Comments and Suggestions for Authors
This is a well-written manuscript that summarizes data from a set of trials done by the CanSino biotech company. They show for the first time that pre-existing adenovirus antibodies do affect the immunogenicity of an inhaled adenoviral SAR-CoV-2 vaccine, which previous publications insisted was not the case.
There are a few issues that need to be addressed. Since the inhaled versions is after a prime with their Ad5 construct, how can we be sure that there was not differential induction of anti-vector antibodies by the vector itself, and if this had been controlled for, then the apparent affect of the pre-existing antibodies may have disappeared. Since the antibodies induced by natural infection and vaccination are not the same (penton for the former and hexon for the latter), this should be addressed in the discussion section.
Also, we need to understand why the adults receiving the inhaled vaccine with high pre-existing antibodies had such a pronounced response (four-fold higher) compared to every other inhaled group. Did this difference disappear when the timing of the boost was taken into account?
Essentially all of the key material is in Figure 1 and I think Tables 2 and 3 could be moved to supplemental material.
There are some spacing and formatting errors throughout, but they are of minor importance.
